# Genetic Variance Estimates for Maize Yield, Grain Moisture, and Stalk Lodging for Doubled-Haploid and Conventional Selfed-Line Hybrids

**DOI:** 10.3390/plants9020138

**Published:** 2020-01-22

**Authors:** Ronald P. Mowers, David J. Foster

**Affiliations:** 1Syngenta Seeds Statistician, 2114 State Ave, Ames, IA 50014, USA; 2Syngenta Seeds, Slater, IA 50244, USA; david.foster@syngenta.com

**Keywords:** doubled haploid, genetic variance, maize yield, selection response

## Abstract

An experiment was conducted to compare estimated genetic variance for maize doubled haploid (DH) with conventional twice-selfed (S2)-line hybrids. Starting with a 4-parent population, at least 160 lines were derived using both of these methods and crossed with two inbred testers. For both inbred testers, maize hybrid grain yield and stalk lodging had higher estimated genetic variances for DH than for S2. For one of the testers, estimated grain moisture genetic variance was higher for DH, but not for the other. The DH hybrid yield distributions on both testers were flatter and had more entries in tails compared with S2 distributions. With complete homozygosity of DH lines and the subsequent increased genetic variance among lines, the expected response to yield selection is higher for DH than for S2 line hybrids.

## 1. Introduction

Seed companies and other organizations such as CIMMYT (International Maize and Wheat Improvement Center) now commonly use doubled haploid (DH) lines to cross with inbred testers to form maize (*Zea mays*, L.) hybrids for testing and selection [1,2,3,4,5]. Geiger and Gordillo [2] cite advantages for DH lines, including complete homozygosity with the maximum genetic variance between lines for hybrid testcross performance, reduced breeding cycle time, increased efficiency of molecular marker applications [3,4], and other efficiencies.

There are theoretical advantages to having all genetic variance exhibited between homozygous DH lines rather than some variance left within lines, as is the case with those derived by selfing [6,7]. In the experiment reported here, we compare the genetic variance between DH-line hybrids with that between conventional S2-line hybrids (derived by selfing twice from a parent population) for empirical evidence on whether there is a difference. Falconer and Mackay [7] (p. 108) partitioned observed phenotypic value P, such as the hybrid yield of an individual, into genotypic value G and environmental deviation E, or P = G + E. When G and E are uncorrelated, total variance is V_P_ = V_G_ + V_E_ (p. 122). The genetic variance V_G_ in our case is variation from hybrid to hybrid after eliminating environmental and genotype-by-environment (G*E) influences, and V_G_ can be estimated. Maize breeders generally select a subset of individual hybrids with best values of, for example, grain yield, to advance for further testing in another season. The true difference of the average of this selected group from the average of all hybrids or of check hybrids is considered the response to selection [7] (pp. 185–190) and can be estimated from the variance components and proportion of upper tail values chosen, a measure of which is called selection intensity. With greater genetic variance and the same proportion of individuals selected, we expect a greater response to selection. Thus, it is helpful to have good estimates of genetic variance for both DH- and S2-line hybrids.

There is a lack of reported estimates of genetic variances comparing maize DH-hybrids to conventionally derived hybrids based on data from actual field experiments, the results of which would be beneficial to scientists and maize breeders. Seitz [1] is one author who reported higher average testcross variance among maize DH-line hybrids than among S2- and S3-line hybrids, but he did not report statistical tests for these differences.

A field experiment was conducted in Garst Seed Company in summer 1997 to compare the hybrid performance of maize doubled haploids with lines obtained by selfing twice (S2) from the same original 4-parent population. The objectives of this experiment were to estimate genetic variances for maize grain yield, grain moisture, and stalk lodging for the sets of DH-line and S2-line hybrids, to validate theory by testing whether DH-line genetic variance in hybrids is greater than that of S2-line hybrids, and to estimate response to selection in the first season of yield trials for the two methods of line derivation. This experiment provides estimates of the genetic variances from which response to selection for maize DH-hybrid yield can be inferred and illustrates DH superiority to conventional S2-line hybrids.

## 2. Results and Discussion

Average values of yield, grain moisture, and stalk lodging for the maize doubled-haploid line hybrids and S2-line hybrids are provided in Table 1. In Table 1, the two inbred testers are designated L and U. Four sets of about 40 hybrids for each method, and using the same tester, were grown in the same five locations. Any differences in means of conventionally S2-derived and DH-line hybrids are insignificant and may have occurred due to confounding field effects because groups of sets were grown in adjacent tiers within the same locations. Overall yield averages for L/DH- and L/S2- line hybrids are 138 and 134 bushels per acre (bu/A), moisture averages are both 17.3%, and stalk lodging averages are 8.9% and 9.7%, respectively. The U/DH- and U/S2-line hybrid averages are also very close to each other, within 0.3 bu/A in yield, 0.3% in grain moisture, and 0.2% in stalk lodging.

Genetic variance estimates are generally greater for DH- than S2-line hybrids (Table 1). The variables statistically analyzed for this difference, using the across-environments degrees of freedom, are grain yield, grain moisture, and the logarithm of (1 + percent stalk lodging), which transforms lodging values to a more normal distribution [8] (pp. 290–291). The 95% confidence intervals for ratios of grain yield genetic variances of DH-line/S2-line crosses were (1.87, 3.45) and (1.27, 2.33), respectively, for the L- and U-testers. These intervals do not contain 1, and we infer that DH hybrids have greater genetic variance than S2 hybrids [8] (pp. 98–99). The yield genetic variance ratio for the L tester, 2.54, was numerically higher than that for the U tester, 1.72, possibly because there was higher average stalk lodging for the L-tester hybrids, and higher stalk lodging can lead to higher yield variance. Our estimates are very similar to the Seitz [1] DH:S2 testcross grain yield ratio 2.14, based on 3 sets (105 total S2 lines), with conventional and DH lines derived from the same source populations and tested in 5 locations in 2003.

For our experiment and the grain moisture trait, the DH:S2 ratio confidence intervals were (0.98, 1.81) and (1.20, 2.21) for the L- and U-testers, respectively. Because the interval for the L tester contains 1, the DH and S2 hybrid genetic variances did not significantly differ (*p* = 0.05), although the DH hybrid genetic variance was numerically greater. For stalk lodging, ratio intervals were (1.10, 2.03) and (1.30, 2.41) for the L- and U-testers, respectively, implying that DH genetic variances are greater than for S2 hybrids.

The greater variances for DH lines imply a potential breeding benefit besides the time savings of deriving doubled haploid lines. Genetic gain is response/time, and apart from decreasing the number of seasons to produce them, DH lines can increase response in the first year of testing, as shown below.

Estimated response R = k*h^2^ *s_p_ for selection intensity k, heritability h^2^, and estimated phenotypic standard deviation s_p_ [7] (p. 189). The heritability h^2^ is a broad-sense heritability on an entry means basis, estimated by s_g_^2^/s_p_^2^., where s_g_^2^ estimates the true genetic variance V_G_ and s_p_^2^ estimates the true phenotypic variance V_P_. For entry means over 5 locations, the phenotypic variance estimate, s_p_^2^ = s_g_^2^ + s_g*e_^2^/5. We can estimate response for our five-location, two-rep test with the maximum likelihood estimates of s_g*e_^2^ and by using the above equation. In units of selection intensity, response is R/k = (genetic variance/ phenotypic standard deviation), estimated as s_g_^2^/s_p_.

Results of the calculations of R/k for first-year testing are contained in Table 2 below. Expected yield responses were 79% higher in DH compared with S2 for L-tester hybrids and 37% higher for U-tester hybrids. Estimated increases in response for DH compared with S2 for grain moisture were 18% in L-tester crosses and 27% in U-tester crosses. Of course, the magnitude of realized responses will not necessarily be this large in a future year because, with a single-year study, we could not include estimated genotype * year variance in the s_p_ formula.

The impact of the increase in genetic variance and expected response for yield and grain moisture are illustrated in the frequency histograms below. The distributions clearly indicate that more superior hybrids are available in the initial year of testing from the set of DH-line hybrids than from the S2-line hybrids. For the L tester, six DH-line hybrids were above the 156 bu/A class vs. none for S2 (Figure 1), and for the U tester, there were five DH-line hybrids above the 168 bu/A class vs. none for S2 (Figure 2). This illustrates a different type of distribution for the DH-line crosses, with fewer observations in the middle and more of them in the tails. Distributions were not as different for grain moisture for the L tester (Figure 3), in concordance with more similar moisture variances (Table 1). For the U tester, there were four observations with 18% grain moisture or less for DH vs. none for S2 (Figure 4).

Another interesting result from this experiment is that the correlation of yields between hybrids of the same line on each of the two testers was higher for the DH lines than for S2 lines. Observed yield correlation between L- and U-tester crosses for S2 lines was 0.37 and for DH lines was 0.61. This may be due to complete homozygosity and less within-line variance giving more consistent estimates of genetic effects for DH lines.

## 3. Materials and Methods

**Field experiment.** The experiment used a 4-parent population from which DH lines and S2 lines were derived. The population started with 4 inbred lines, A, B, C, and D, which were crossed in pairs. Next, these F1 pairs were crossed with each other to get three sub-populations: (A × B) × (C × D), (A × C) × (B × D), and (A × D) × (B × C), which were blended together in a balanced bulk. Lines were derived from these 4-parent populations in about equal numbers for both methods (DH and S2). In the conventional method (S2), there were two generations of selfing, once in a nursery in Iowa, USA, and once in Hawaii. For the DH method, induction of maternal haploids and artificial chromosome doubling, as described by Geiger and Gordillo [2], was used.

The two sets of derived lines were crossed with each of two male tester inbreds, called L and U, to form four groups of hybrids: two testers, and two line-derivation methods (DH and S2). There were approximately 160 hybrids per group, and these were divided equally among four yield trial sets for the purpose of hybrid testing. Keeping hybrids of the same group within each of the sets was done to ensure conditions were as similar as possible to those of future yield trials, namely hybrids with the same tester, same line-derivation method (DH-line or S2), and evaluated in sets of about 40 or 50 hybrids. This experiment design does not allow good mean yield comparisons between line-derivation methods, but it does lead to providing variance estimates without having to contend with plot-to-plot competition effects from mixing line derivation methods.

Each yield trial set was grown in a randomized complete block design with 2 reps. Hybrids from both derivation methods involving a given tester were grown at the same five locations in the Midwest US. The two testers differed in maturity, so the sets of hybrids on the earlier tester were not grown at all the same locations where the later maturing sets of hybrids were grown, but there were two locations in common to both testers. The S2 hybrids were grown in 50-entry sets, with 40 derived-line hybrids and 10 checks. DH hybrids were grown in 45-entry sets with 38 or 39 derived-line hybrids and 7 or 6 checks. For a given tester, six check hybrids were common to the DH and S2 sets. Two checks were common to all sets across both testers.

Variables measured on each two-row, 1.5 × 6 m field plot of the yield trial experiments are final plant stand; grain yield corrected to 15.5% grain moisture, measured in bushels per acre (bu/A) and converted to metric tons per hectare by multiplying by 0.06278; grain moisture percentage at harvest; and stalk and root lodging and dropped ear counts, each expressed as a percentage of final stand. Too few plots had root lodging and dropped ears to report results here.

**Statistical analyses.** Analyses were done for the traits grain yield, grain moisture, and stalk lodging to get hybrid least-squares means (LSmeans) over reps, estimate genetic variances using analysis of variance and restricted maximum likelihood (REML) [9] (pp 498–9), get confidence intervals for ratios of variances, estimate response to selection, and correlate yields for lines crossed with both testers. SAS [10] software was used.

Analysis of variance was done for each of the testers by derivation method groups (L-DH, L-S2, U-DH, U-S2). Each group had 20 RCBD experiments (5 locations with 4 sets of hybrids each) analyzed with the model Y = Rep Hyb; to compute hybrid LSmeans for each of the traits measured. In the balanced data case, these are averages over reps. As example of the output, the U-S2 group had 50 hybrids in each set: 10 checks and 40 U/S2-line hybrids; from this analysis, there were 50 hybrid LSmeans for each set for a total of 200 hybrid means for each location. The 20 independent experiments for each tester-method group are considered 20 set-within-location environments. For each, we equated expected mean squares to observed values to solve for estimated genetic variances, then averaged these 20 estimates to get an estimate of genetic variance for that tester-method group.

We also estimated genetic variance components over all locations and sets using SAS Proc Mixed [10] and REML [9]. For example, the data set for U/S2-line hybrids had 1000 total means (5 locations * 200 means per location). Using Proc Mixed; the model was Y = Loc + Set(Loc) + Chk + Chk*Set(Loc); with random hybrid to estimate genetic variance for derived line hybrids. The residual mean square from this mixed model is the genotype * environment (G*E) variance. For the U-S2 group, we had fixed effects for locations (5), sets within locations (20), check hybrids (10), and check hybrid interactions with locations and with sets within locations. The random factor is S2-line hybrids (total of 159, because one missing), which gives another estimate of genetic variance, in almost complete agreement with the analysis of variance method estimate. The residual had 620 degrees of freedom to estimate G*E variance.

Confidence intervals for ratios of DH-line to S2-line hybrid genetic variances were calculated. Snedecor and Cochran [8] (p. 98) discuss the test of null hypothesis of equality of two variances using the ratio of estimated variances (s_1_^2^/s_2_^2^), distributed as F with d_1_ and d_2_ numerator and denominator degrees of freedom. A 95% confidence interval for this variance ratio is
(s_1_^2^/s_2_^2^)/F(0.025,d_1_,d_2_) < (σ_1_^2^/σ_2_^2^) < (s_1_^2^/s_2_^2^)/F(0.975,d_1_,d_2_),
where *F(p,d_1_,d_2_)* is F-value for upper-tail probability *p*. These confidence intervals are based on normal distributions, a reasonable assumption for both grain yield and moisture. However, percent stalk lodged plants had linear relation of standard deviations with means and was transformed [8] (p. 290) to log(percent stalk lodged + 1) to better estimate genetic variance ratios and confidence intervals. Degrees of freedom *d_1_* and *d_2_* were 20, the numbers of environments contained in each group.

Response was estimated for traits using the formula on page 189 of Falconer and McKay [7], R = kh^2^σ_P_, where *k* is intensity of selection, *h^2^* is broad-sense heritability on an entry-means basis, and *σ_P_* is phenotypic standard deviation on an entry-means basis. This formula and estimates of genetic variance allowed estimation of response to selection for each of the line-derivation methods. Correlations of yields of hybrids with the same line on each tester were done for the DH lines and for S2 lines. We combined data over sets and locations to get LSmeans for each hybrid in each tester-method group. Means from the L- and U-testers were paired, matched on the derived line. For each derivation method, correlation of L/line with U/line hybrids was done to indicate consistency of the line performance over the two testers: correlation 1 indicating the exact same spacing and ordering of lines for both testers and correlation 0 indicating a random ordering for the testers.

## 4. Conclusions

This experiment was designed to compare the performance of maize hybrids made from DH lines with hybrids using conventional S2-lines. A common 4-parent population was the breeding source for the lines created with the two derivation methods, DH and S2. All derived lines were crossed with two inbred testers, and about 160 hybrids were grown in 5 locations for each tester by line-derivation method combination.

The main difference in DH- and S2-line hybrid performance was in estimated genetic variances. For both inbred testers, maize hybrid grain yield and stalk lodging had significantly (*p* = 0.05) higher estimated genetic variances for the DH method than for the S2 method. For one of the inbred testers, estimated yield genetic variance was 2.54 as large for DH as for conventional S2-derived lines, and for the other tester 1.72 times as large. Seitz [1] had found a ratio of DH:S2 hybrid yield genetic variance of 2.14, similar to our estimates. Grain moisture genetic variance for one of the testers was significantly higher for DH hybrids than for S2, but not for the other tester.

Higher genetic variance for DH lines results in a change in the shape of distributions, with fewer maize hybrids in the middle and more in the tails of the DH-hybrid distributions. Expected response from first-year yield trials is, therefore, greater for DH-line hybrids than for conventional S2-line hybrids.

This work provides evidence of positive differences in hybrid performance in yield trials for DH lines versus S2 lines derived from the same 4-parent population. Results from this experiment demonstrate empirically the theoretical breeding advantage of a higher genetic variance associated with DH lines contrasted to lines derived from conventional selfing.

## Figures and Tables

**Figure 1 plants-09-00138-f001:**
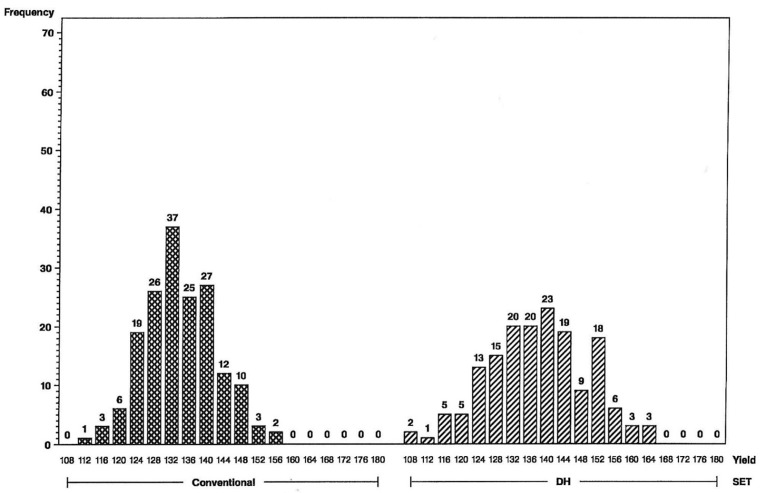
Frequency distributions for maize hybrid grain yields in bushels per acre (on *x*-axis) using the L inbred tester, conventional S2 line-hybrids on the left and DH line-hybrids on the right. Numbers of hybrids are above the bars.

**Figure 2 plants-09-00138-f002:**
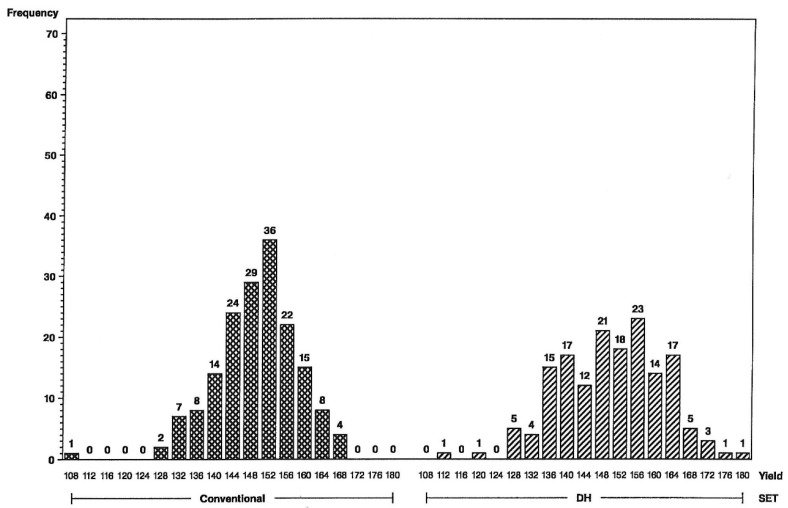
Frequency distributions for maize hybrid grain yields in bushels per acre (on *x*-axis) using the U inbred tester, conventional S2 line-hybrids on the left and DH line-hybrids on the right. Numbers of hybrids are above the bars.

**Figure 3 plants-09-00138-f003:**
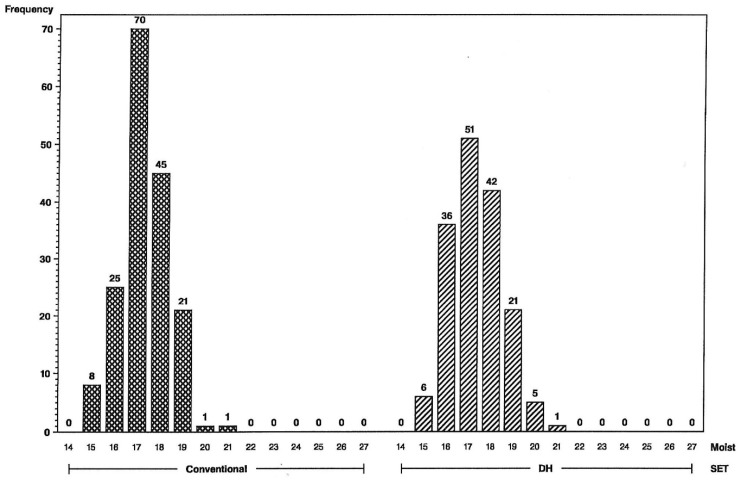
Frequency distributions for maize hybrid grain moisture in percent (on *x*-axis) using the L inbred tester, conventional S2 line-hybrids on the left and DH line-hybrids on the right. Numbers of hybrids are above the bars.

**Figure 4 plants-09-00138-f004:**
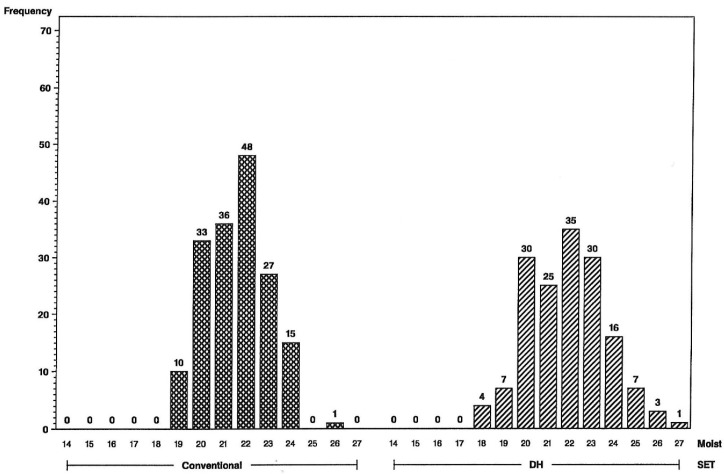
Frequency distributions for maize hybrid grain moisture in percent (on *x*-axis) using the U inbred tester, conventional S2 line-hybrids on the left and DH line-hybrids on the right. Numbers of hybrids are above the bars.

**Table 1 plants-09-00138-t001:** Averages and genetic variance estimates for yield, grain moisture, and percent stalk lodging for each tester and both methods of line derivation.

Tester	Method	Yield			Grain Moisture			Stalk Lodging		
		Mean	Genetic Variance	DH/S2 Var	Mean	Genetic Variance	DH/S2 Var	Mean	Genetic Variance	DH/S2 Var
**L**	**DH**	137.5	106.08	2.54	17.30	1.081	1.33	8.93	13.25	1.47
**L**	**S2**	133.8	41.75		17.28	0.812		9.70	9.02	
**U**	**DH**	149.3	109.00	1.72	21.85	2.571	1.63	3.55	3.10	2.18
**U**	**S2**	149.0	63.40		21.53	1.581		3.40	1.42	

**Table 2 plants-09-00138-t002:** Estimates of variances and response for yield and grain moisture, DH vs. S2 hybrids with the L and U testers, data from five Midwest US locations in 1997.

	Yield: L-Hybrids		Yield: U-Hybrids		Moisture: L-Hybrids		Moisture: U-Hybrids	
	DH	S2	DH	S2	DH	S2	DH	S2
**s_g_^2^**	106.1	41.8	109	63.4	1.08	0.81	2.57	1.58
**s_g*e_^2^**	150.2	128.8	133.3	116.7	0.64	0.66	1.13	0.66
**R/k**	9.09	5.09	9.36	6.81	0.98	0.83	1.54	1.21

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
