# Peer review of "Genetic Variance Estimates for Maize Yield, Grain Moisture, and Stalk Lodging for Doubled-Haploid and Conventional Selfed-Line Hybrids"

_plants, 2020, doi:10.3390/plants9020138_

Round 1
Reviewer 1 Report
The work provide interesting information about the use of DH and S2 lines in maize breeding.
I would suggest to improve the way the manuscript is presented and in particular to follow these comments.
The introduction can be improved with more information and with providing references that are easy to find. In addition the use of bulleted list should be avoided.
Conclusion contain a summary of the work done, and a very brief comment. It should substantiate the achieved results with more strength and putting them in relation with previous work on the same subject.
Author Response
Thank you for very good comments. I have tried to improve the Introduction by including more information and providing a couple more references which are easy to find. However, the key reference (Seitz) is not in a refereed journal, making it difficult to find except at a library contaioning Illinois Corn Breeder School proceedings. I tried to summarize the important information relative to our experiment in the results and discussion and in the summary. I also avoided the bulleted list in the Introduction.
For the conclusion, I tried to substantiate achieved results with more strength and put them in relation to previous work (Seitz, 2005) on this subject.
Throughout the revised manuscript, we attempted to improve wording and explain technical points more in a layman's terms. Statistical analysis methods were added to more adequately describe methods. We attempted to more clearly present results.
Again, thank you for helping us get a (hopefully) much improved revision to the original manuscript.
Reviewer 2 Report
This manuscript provides field data that tests (and supports) the improved utility of doubled-haploid-derived hybrids versus conventionally-selfed-line-derived hybrids in test crosses for estimating genetic variance and selection response. The experimental design is well-conceived and clearly presented. Authors demonstrate larger genetic variance scores for populations derived from doubled-haploids crossed with testers compared to selfed hybrids crossed with the same testers. Yield, grain moisture, and stalk lodging, all important traits for maize breeders, are the traits analyzed. The manuscript demonstrates that the DH lines, besides being faster to produce, also provided better material for obtaining response to selection measurements in test crosses, having substantially larger estimated genetic variances and all comparisons.
While well-written, the language is very technical and assumes expertise in the field of maize breeding. I would suggest providing additional detail behind some of the terms and assumptions to help readers with less familiarity in the area. Discuss the genetic and theoretical reasons why it is expected that DH would outperform S2-derived lines in estimated genetic variance and why this is important. Also define the following in layman’s terms: Estimated Genetic Variance, Expected Response and Response to Selection, Selection Intensity, Expected Yield Response.
On Line 42: differences are inconsequential or insignificant?
Author Response
Thank you for very good comments. They help tremendously.
Yield, grain moisture, and stalk lodging are important traits for maize breeding, and DH-line hybrids in our experiment led to larger estimated genetic variances compared with S2-line hybrids.
I have tried to improve the Introduction by including a couple more references. The key reference (Seitz) is not in a refereed journal, making it difficult to find except at a library containing Illinois Corn Breeder School proceedings. I tried to summarize the important information relative to our experiment in the results and discussion and in the summary. The technical terms such as genetic variance, response to selection, and selection intensity were addressed in the Introduction with a short summary of Falconer and Mackay's presentation of the terms, hopefully to simplify them.
We tried to provide additional detail behind technical terms and assumptions. (There is some heavy statistical and maize breeding technology here.) We attempted to discuss theoretical genetic reasons why DH should out-perform S2 for genetic variance and why this is important.
For the conclusion, I tried to substantiate achieved results with more strength and put them in relation to previous work (Seitz, 2005) on this subject.
Throughout the revised manuscript, we attempted to improve wording and explain technical points more in a layman's terms. Statistical analysis methods were added to more adequately describe methods. We attempted to more clearly present results.
Again, thank you for helping us get a (hopefully) much improved revision to the original manuscript.